# A fluorinated cation introduces new interphasial chemistries to enable high-voltage lithium metal batteries

Qian Liu[1], Wei Jiang[2], Jiayi Xu[1], Yaobin Xu [3], Zhenzhen Yang[1], Dong-Joo Yoo[1], Krzysztof Z. Pupek[4], Chongmin Wang [3], Cong Liu [1], Kang Xu [5] ✉ & Zhengcheng Zhang [1] ✉

Fluorides have been identified as a key ingredient in interphases supporting aggressive battery chemistries. While the precursor for these fluorides must be pre-stored in electrolyte components and only delivered at extreme potentials, the chemical source of fluorine so far has been confined to either negatively-charge anions or fluorinated molecules, whose presence in the inner-Helmholtz layer of electrodes, and consequently their contribution to the interphasial chemistry, is restricted. To pre-store fluorine source on positive-charged species, here we show a cation that carries fluorine in its structure is synthesized and its contribution to interphasial chemistry is explored for the very first time. An electrolyte carrying fluorine in both cation and anion brings unprecedented interphasial chemistries that translate into superior battery performance of a lithium-metal battery, including high Coulombic efficiency of up to 99.98%, and $Li^0$-dendrite prevention for 900 hours. The significance of this fluorinated cation undoubtedly extends to other advanced battery systems beyond lithium, all of which universally require kinetic protection of highly fluorinated interphases.

The lithium-ion battery (LIB) is not only the most popular electro-chemical device invented by mankind, but it is also the very first battery relying on interphases, because its electrode materials (graphitic anodes and transition metal oxide cathodes) must operate at potentials far beyond the thermodynamic stability limits of any known electrolytes[1,2]. The excellent reversibility of modern LIBs, as characterized by thousands of cycles and up to hundred years of calendar life[3], is ensured by the interphases that are formed by the sacrificial decompositions of electrolyte components during the initial activation cycles. Although the exact formation mechanism and structure of the interphases remain to be understood, persistent studies on this important sub-component in the past decades have identified the electrolyte solvents (carbonate esters) and salt anions (hexafluorophosphate $PF_6^-$ or tetrafluoroborate $BF_4^-$) as the major chemical sources for the interphasial chemistry[4–9]. More recent investigations have revealed that the fluorides from those anions and solvents might serve as key ingredients in resisting electrochemical reductions and oxidations while providing fast $Li^+$-conduction pathways via interfacing with semi-carbonates at nano-length scale[10–13]. Therefore, designing an interphase with enriched, but nanosized fluorides that are evenly distributed in the heterogenous matrices of carbonates, oxides, and polymeric species becomes the central mission of developing better electrolytes for future battery chemistries of high energy densities. Such an interphase would dictate the success of "Holy Grail" batteries based on lithium-metal anode and high capacity, high voltage cathodes, as well as other advanced battery concepts beyond lithium-based chemistries.

[1]Chemical Sciences and Engineering Division, Argonne National Laboratory, Lemont, IL 60439, USA. [2]Computational Science Division, Argonne National Laboratory, Lemont, IL 60439, USA. [3]Environmental Molecular Sciences Laboratory, Pacific Northwest National Laboratory, Richland, WA 99352, USA. [4]Applied Materials Division, Argonne National Laboratory, Lemont, IL 60439, USA. [5]Battery Science Branch, Energy Science Division, Sensor and Electron Devices Directorate, U.S. Army Research Laboratory, Adelphi, MD 20783, USA. ✉e-mail: conrad.k.xu.civ@army.mil; zzhang@anl.gov

Despite the various efforts of fluorinating the interphases, including the most recent super-concentrated[14–16] or locally-concentrated electrolytes[17–23] as well as all-fluorinated electrolytes[24–28], the chemical sources of these interphasial fluorine formed in all the investigated electrolytes have been restricted to two classes of species: (1) the fluorinated salt anions ($PF_6^-$, $BF_4^-$, or the newly developed bis(trifluoromethanesulfonyl)imide, $TFSI^-$, or bis(fluorosulfonyl)imide, $FSI^-$) that bear negative charge and (2) fluorinated solvent molecules that bear no charge. This restricted source brings an intrinsic disadvantage to the desired chemistry of interphases, because these fluorinated precursors could not populate the inner-Helmholtz layers of the electrode surface of a high Fermi energy level, such as $Li^0$ anode, hence their participation in the interphasial chemistry thereon was consequently handicapped. Such a disadvantage can be most visibly evidenced in the so-called "cathodic challenge" encountered in the efforts of protecting anode surfaces in aqueous electrolytes, because in that case the interphasial chemistry would completely rely on the contributions from anion reduction[29–31]. Beside the charge of these fluorine precursors, the nature of the fluorine bonds also matters, as the C−F bonds seem to provide higher quality interphasial fluorination than the labile fluorines sitting on heteroatoms, such as phosphorus, boron, or sulfur[32,33]. Hence, in the broader context, a cation bearing C−F bonds is a highly coveted structure, because its presence, along with fluorinated anions and solvent molecules, would ensure maximum and balanced fluorination of interphases on both anode and cathode surfaces.

In this work, we successfully synthesized a fluorinated cation, 1-methyl-1-propyl-3-fluoropyrrolidinium and apply this F-cation concept on interphase in high-voltage Li-metal batteries for the first time. When coupled with a fluorinated anion $FSI^-$, it forms an ionic liquid (PMpyr$_f$FSI) that has zero vapor pressure, has complete non-flammability, and carries potential fluorine sources on both cation and anion. Lithium salt with the same anion (LiFSI) can dissolve in this ionic liquid at various concentrations, leading to an ionic liquid electrolyte that offers unprecedented interphasial chemistry opportunities when the electrolyte interacts with both cathode and anode of the two extreme potentials.

## Results and discussion

### Electrolyte stability

PMpyr$_f$FSI containing one fluorine in the pyrrolidinium cation ring was synthesized through a one-step quaternization method with methyl bis(fluorosulfonyl)imide (MeFSI) as the methylating agent with 100% conversion (Fig. 1a, b). Density function theory (DFT) calculations reveal that, in comparison with non-fluorinated counterpart, 1-methyl-1-propylpyrrolidinium (PMpyr$^+$), the presence of fluorine reduces the energy levels of both HOMO (from −13.18 to −13.29 eV) and LUMO (from −3.47 to −3.66 eV) of the cation (Fig. 1a). Such down shift in HOMO/LUMO energy indicates the increased resistance of the cation against oxidation but also shows lowered resistance against reduction[34]. Consistent with HOMO energy calculation by DFT, PMpyr$_f$FSI shows good anodic stability up to 5.5 V vs. Li$^+$/Li in cyclic voltammetry test, which is slightly higher than the 5.4 V for PMpyrFSI[35]. Moreover, the higher oxidation stability of PMpyr$_f$FSI is even manifest from the potentiostatic hold experiment conducted at higher voltages from 4.6 to 4.9 V, as it exhibits lower leakage current compared to PMpyrFSI (Supplementary Fig. S1).

Potential energy surfaces were also constructed based on DFT calculations. The pyrrolidinium cation, either with or without the fluorine, binds equally strong to Li$^0$ surface (−2.12 and −2.10 eV, respectively), but much stronger than the $FSI^-$ anion (−0.66 eV). However, PMpyr$_f^+$ undergoes reductive decomposition with C−F cleavage, which is exothermic by −3.67 eV, as compared with C−H cleavage that is exothermic by −0.26 eV, followed by the ring-opening process, which is exothermic by −0.87 eV (Fig. 1c). Therefore, the introduction of fluorine renders the reduction of PMpyr$_f^+$ much more favored than PMpyr$^+$, while such difference in reactivity is primarily due to the strong interaction between the F and Li (110) surface. As indicated by the density of states (Supplementary Fig. S3), the significant orbital overlapping between Li-p orbital and F−s and p

**a** HOMO/LUMO change of fluorinated IL

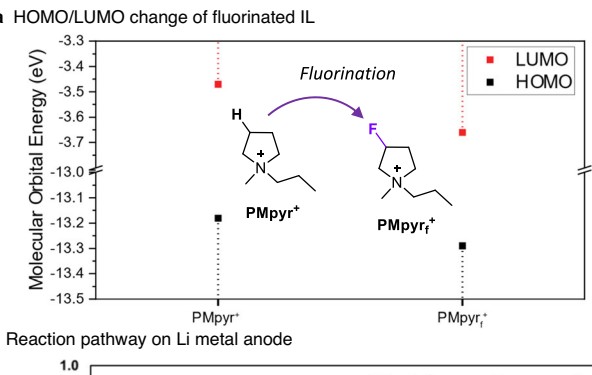

**b** Synthesis of PMpyr$_f$FSI

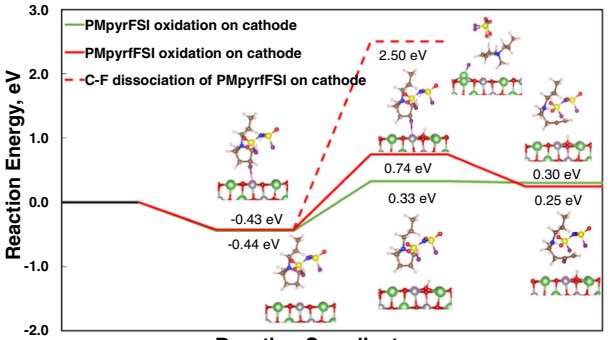

**c** Reaction pathway on Li metal anode

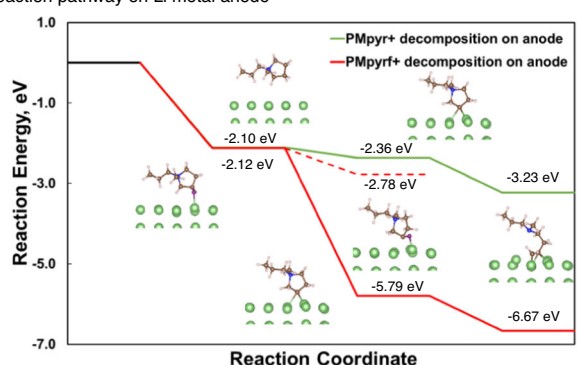

**d** Reaction pathway on NMC622 cathode

**Fig. 1 | Synthesis and DFT calculations of fluorine substituted cation PMpyr$_f^+$. a** Comparison of HOMO/LUMO energy levels for PMpyr$_f^+$ and PMpyr$^+$. **b** Synthesis route for PMpyr$_f$FSI. **c** PMpyr$_f^+$ and PMpyr$^+$ reduction pathway on Li metal. **d** PMpyr$_f^+$ and PMpyr$^+$ oxidation pathway on NMC622 cathode.

orbitals indicates the strong interaction. However, the orbital overlapping cannot be found for H interacting with the Li (110) surface. The strong interaction between F and Li (110) surfaces makes C−F cleavage more energetically favorable than C−H cleavage by −3.41 eV, which would preferentially generate inorganic or organic fluorides. Deprotonation pathway was also considered for the $PMpyr_f^+$ cation decomposition, this mechanism is exothermic by −0.66 eV (Fig. 1c, red dash line). In comparison with $PMpyr^+$ cation, the deprotonation becomes more thermodynamically favored (−0.66 vs. −0.26 eV), which could be due to the presence of electron-withdrawing F that weakens the C−H bond that shares the same carbon. This demonstrated that deprotonation could also participate in SEI formation, however, the defluorination is the dominating pathway. Meanwhile, the $FSI^-$ decomposition is also highly likely (Supplementary Fig. S2b), as evidenced by the exothermic S−F bond cleavage of −5.97 eV, followed by exothermic dissociation of S−O bond, which is −2.94 eV. The fluorides thus generated from the labile bonds in $FSI^-$ would more likely exist in inorganic forms with heteroatom contents (P, S) and behave less protectively as the fluorides generated from C−F bonds[36]. To mimic the decomposition during charge state, the potential energy surface with consideration of $e^-$ transfer after adsorption is constructed and shown in Supplementary Fig. S2a, S2b. The reaction energy for the $FSI^-$, $PMpyr_f^+$, and $PMpyr^+$ activations are −6.82, −6.55, and −3.14 eV, respectively. Thus, the corresponding reduction potentials are 5.43, 5.16, and 1.75 V, according to Eq. (1) in "Methods". In a nutshell, the cations bind to the Li (110) surface much stronger than anions do; however, without F-substitution, $PMpyr^+$ are not as active as $FSI^-$ toward reduction. With F-substitution, $PMpyr_f^+$ is more active towards reduction in comparison with $PMpyr^+$ and is almost as active as $FSI^-$ anion. Fortunately, the higher adsorption preference of $PMpyr_f^+$ over $FSI^-$ should lead to more abundant fluorides from C−F origin−due to the fluorinated cation.

On the NMC622 cathode side, PMpyrFSI and $PMpyr_f$FSI bind equally strong with Mn sites through the cation part with cation adsorption energy of 0.44 eV and 0.43 eV, respectively; however, the adsorption is much weaker when binding through anion part (0.39 eV). Therefore, we will only consider cation oxidation during the charging cycle. As indicated by Fig. 1d, $PMpyr_f$FSI oxidation favors a deprotonation rather than a C−F dissociation pathway, due to the lower potential energy for deprotonation compared with C−F cleavage (1.17 eV vs 2.93 eV). Furthermore, the C−N bond is weakened, followed by a ring-opening process of deprotonated $PMpyr_f$FSI, which results in a potential energy of −0.49 eV. However, in the absence of fluorine, PMpyrFSI tends to be more easily oxidized with reaction potential energy of 0.77 eV for the deprotonation step, followed by the exothermic ring-opening process of −0.03 eV. This indicates that F-substitution indeed renders $PMpyr_f$FSI more resistant against oxidation, as predicted by the downshift of HOMO/LUMO energy. Apparently, what was brought by fluorination is the improvement in thermodynamic stability against oxidation on cathode surfaces, as well as a strong tendency to decompose on the $Li^0$ anode. As we have learned from the knowledge in LIB electrolytes, the latter strongly implies the preferential cation reduction to form interphases on $Li^0$.

An ionic electrolyte $(PMpyr_f)_{0.8}Li_{0.2}FSI$ was formulated by dissolving LiFSI in $PMpyr_f$FSI, whose performance was evaluated in a lithium-metal cell consisting of NMC622 cathode and a thin (20 μm) lithium anode. Molecular dynamics (MD) simulation sheds light on the electrode/electrolyte interfacial structure in such batteries. Compared with its non-fluorinated cousin, the fluorine presence on the cation (purple) makes it much easier to access and populate the inner-Helmholtz layers on both the NMC622 cathode and Li-metal anode (Fig. 2a, b), which suggests stronger surface interaction of $PMpyr_f^+$ cation with those electrode surfaces and predicts an interphasial chemistry with higher participation from $PMpyr_f^+$. The same insight was provided from the static distribution of the electrolyte

components on both electrodes (Fig. 2d, e). On both $Li^0$ and NMC622 cathode surfaces, the $PMpyr_f^+$ cations are much more populated within the inner-Helmholtz layer than its counterpart $PMpyr^+$ without fluorine.

## Performance in battery

Although fluorination of the cation favors the desired interfacial structure and interphasial chemistry of high fluorine abundance, excess $PMpyr_f^+$ content in the electrolyte does increase bulk viscosity and leads to lower ion transport and higher interfacial impedance (Supplementary Fig. S4). Therefore, we must further optimize the electrolyte formulation by considering the impact of interfacial impedance (Supplementary Fig. S9a). Different from neutral organic solvent-based electrolytes, where $Li^+$ ion is solvated by both solvent molecules and Li salt anion, $Li^+$ ion in pure ionic liquid-based electrolytes is solely solvated by anions and ionic liquid cation does not participate in $Li^+$ ion solvation. Optimization of the $PMpyr_f^+$ concentration in the inner-Helmholtz layer can be achieved by adjusting $Li^+$ concentration in the format of lithium salt. The $(PMpyr_f)_{0.5}Li_{0.5}FSI$ electrolyte is the composition that better balances these considerations. At this composition, the interfacial structure, as well as the static distribution of the electrolyte components, becomes less adsorbed on both electrode surfaces (Fig. 2c, f). Instead, more $Li^+$ (green) are observed on both electrode surfaces, and the purple F-group from $PMpyr_f^+$ cation is reduced, especially on the Li-metal surface.

The electrolyte interfacial structure alteration causes corresponding changes in solid−electrolyte−interphase (SEI) and cathode−electrolyte−interphase (CEI) chemistries, which are revealed by X-ray photoelectron spectroscopy (XPS) analysis. On a Li-metal surface (Fig. 3a), peaks observed in F $1s$ spectra are assigned to be $SO_2F$ at 687.8 eV, C−F at 686.5 eV and LiF at 684.5 eV, where $SO_2F$ is contributed from $FSI^-$ decomposition, C−F is contributed from $PMpyr_f^+$ cation, and LiF is contributed from both $FSI^-$ decomposition and $PMpyr_f^+$ cation defluorination. Peaks observed in N $1s$ spectra are assigned to be $N-SO_2F$ ($FSI^-$ decomposition) at 400.6 eV, $C-N^+$ ($PMpyr_f^+$ cation decomposition) at 402.3 eV and $LiNC_xH_y$ ($PMpyr_f^+$ cation decomposition via ring-opening mechanism) at 399.2 eV. At decreased $PMpyr_f^+$ concentration, the peak relative intensity of 402.3 eV is decreased and 399.2 eV is increased in N $1s$ spectra for $(PMpyr_f)_{0.5}Li_{0.5}FSI$ electrolyte, suggesting $PMpyr_f^+$ contribution to the SEI is mainly through the ring-opening mechanism, which also contributes to higher LiF content as observed in F $1s$ spectra. Moreover, lower C atomic concentration and higher S atomic concentration are observed in $(PMpyr_f)_{0.5}Li_{0.5}FSI$ electrolyte, suggesting reduced $PMpyr_f^+$ cation decomposition on Li metal, as C should have solely come from the cation and S from the $FSI^-$ anion, respectively. This shift of interphasial chemistry between $PMpyr_f^+$ cation and $FSI^-$ anion is apparently a result of $PMpyr_f^+$ cation reduction in interfacial region. On NMC622 cathode, a similar trend is observed, in which the intensities of the C−F peak and C−N peaks contributed from $PMpyr_f^+$ cation decomposition are relatively lower. A peak at 529.2 eV is assigned to metal oxide (M−O) species in O $1s$ spectrum for $(PMpyr_f)_{0.5}Li_{0.5}FSI$ electrolyte, suggesting that the cathode surface is visible now through the formation of a thinner CEI layer (Supplementary Fig. S8). Supporting this argument, both N and S are also observed in lower atomic concentration because of less cation and anion decomposition overall. While both $PMpyr_f^+$ cation and $FSI^-$ anion participate in the SEI formation on the Li-metal anode and CEI formation on NMC622 cathode, the contribution between cation and anion needs to be balanced to enable optimized interphaisal chemistry, which can be achieved by adjusting the cation concentration, and the impact of $PMpyr_f^+$ cation concentration is more significant on Li metal due to the stronger Coulombic interaction.

The $(PMpyr_f)_{0.5}Li_{0.5}FSI$ electrolyte was then tested in a NMC622/Li-20 μm cell (designated as full cell hereafter) cycled between 4.6 and

 

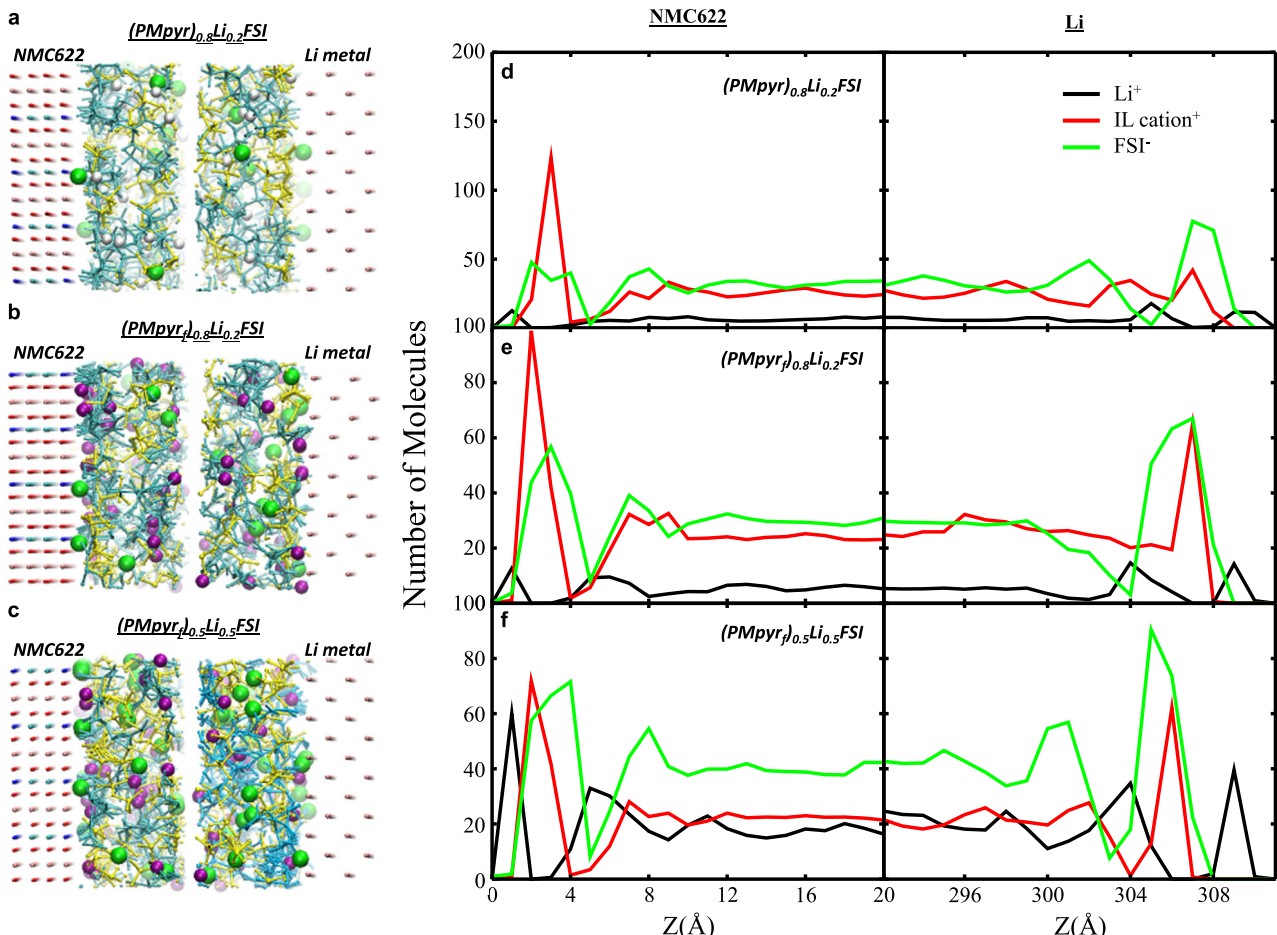

**Fig. 2 | MD simulation on electrolyte/electrode interaction. a–c** Snapshot of electrolyte distribution on NMC622 and Li electrodes: cyan-IL cation⁺; white-H on the PMpyr⁺ backbone highlight in Fig. 1b; purple-F on the PMpyr_f⁺ backbone; yellow-FSI⁻; green-Li⁺. **d–f** Molecular number density profiles along the z-axis normal to the surface of the NMC cathode (left panels) and lithium anode (right panels). The center of mass of each molecule is used to calculate molecule distribution: **a, d** $(PMpyr)_{0.8}Li_{0.2}FSI$ electrolyte, **b, e** $(PMpyr_f)_{0.8}Li_{0.2}FSI$ electrolyte and **c, f** $(PMpyr_f)_{0.5}Li_{0.5}FSI$ electrolyte.

3.0 V (N/P ratio is around 2). Despite its higher room-temperature viscosity (528 mPa s)[35], $(PMpyr_f)_{0.5}Li_{0.5}FSI$ delivers a much higher initial discharge capacity of 203 mAh/g at C/3, as compared to 147 mAh/g for $(PMpyr_f)_{0.8}Li_{0.2}FSI$ (Fig. 4a). This capacity was well-retained, with only 3 mAh/g lost during the first 100 cycles, at an average coulombic efficiency (CE) of 99.9%. The rate capability for $(PMpyr_f)_{0.5}Li_{0.5}FSI$ electrolyte is also superior to $(PMpyr_f)_{0.8}Li_{0.2}FSI$ electrolyte, as evaluated in NMC622/Li-450 μm cell (designated as half-cell hereafter) with 4.6 V upper cutoff voltage (UCV). Although the impedance contributed from $(PMpyr_f)_{0.5}Li_{0.5}FSI$ is around 57 Ω cm², which is much higher than 16 Ω cm² for $(PMpyr_f)_{0.8}Li_{0.2}FSI$ due to the viscosity impact, the overall surface impedance is only 85 Ω cm² for $(PMpyr_f)_{0.5}Li_{0.5}FSI$, substantially lower than 231 Ω cm² for $(PMpyr_f)_{0.8}Li_{0.2}FSI$ after three formation cycles (Supplementary Fig. S9a) and consistent with the observed cycling and rate performance. The enhanced cycling and rate performance and lower surface impedance are attributed to the balanced cation and anion contributions to the CEI and SEI chemistries. The full-cell cycling with $(PMpyr_f)_{0.5}Li_{0.5}FSI$ electrolyte was further extended to 300 cycles (Fig. 4c). High-capacity retention of 89% is achieved for 300 cycles, with minimum overpotential buildup (Supplementary Fig. S9e). After 300 cycles, the surface impedance in $(PMpyr_f)_{0.5}Li_{0.5}FSI$ electrolyte cell is only slightly increased (Supplementary Fig. S9b), in contrast with the conventional electrolyte Gen 2 (1.2 M LiPF₆ in EC/EMC 3/7 weight ratio), which could initially deliver 220 mAh/g capacity at C/3 rate due to its lower viscosity and high conductivity (7.5 mS/cm at 25 °C), but rapidly fades due to its low

oxidation stability on the cathode and the poor stability towards the Li-metal anode.

The half-cell performance at 4.6 V was also evaluated with Gen 2 and $(PMpyr_f)_{0.5}Li_{0.5}FSI$ electrolyte (Supplementary Fig. S10a). By supplying the cell with an unlimited Li source, we can examine the electrolytes high-voltage performance without the Li-metal factor. The cycling performance of Gen 2 is much improved under such a circumstance, with similar fading patterns to the first 30 cycles in the Gen 2 full-cell performance, which is mainly due to the cathode-side degradation. Again, a stable cycling is achieved by $(PMpyr_f)_{0.5}Li_{0.5}FSI$ electrolyte in the half-cell. These results suggest that a good NMC622 surface protection is enabled by $(PMpyr_f)_{0.5}Li_{0.5}FSI$ electrolyte. The nearly identical cycling performance for $(PMpyr_f)_{0.5}Li_{0.5}FSI$ electrolyte in both half and full-cells within 100 cycles also indicates its high Li-metal compatibility. The $(PMpyr_f)_{0.5}Li_{0.5}FSI$ electrolyte also demonstrates stable cycling performance under different UCV, as shown in Supplementary Fig. S9b–S9d. The great cyclability of $(PMpyr_f)_{0.5}Li_{0.5}FSI$ electrolyte compared to the conventional electrolyte suggests this electrolyte has high-voltage stability on the Ni-rich cathode and high Li-metal compatibility. More importantly, the results indicate the modified CEI and SEI are highly protective against further electrolyte degradation over long-term cycling.

## Stabilization of Li metal

The high stability of $(PMpyr_f)_{0.5}Li_{0.5}FSI$ toward Li metal was further confirmed in Li/Cu cells and Li/Li symmetric cells. The CE of Li-metal

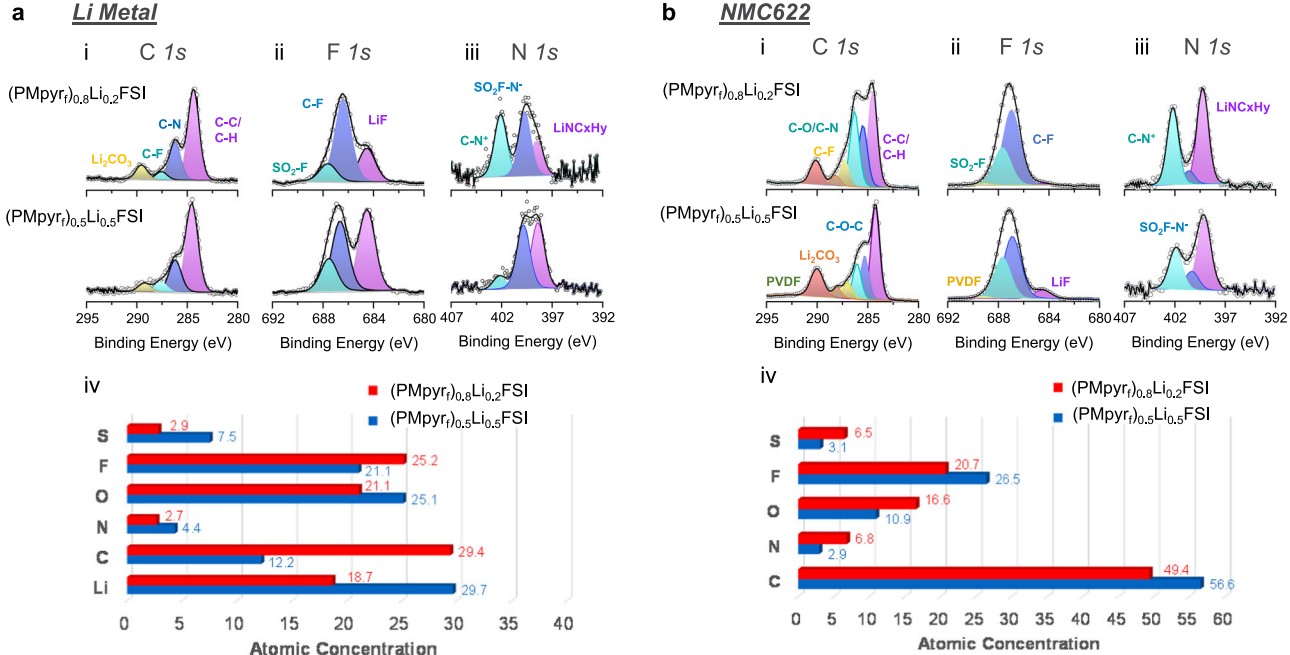

**Fig. 3 | XPS analysis of recovered electrodes. a** Li-metal anode and **b** NMC622 cathode harvested after formation using $(PMpyr_f)_{0.8}Li_{0.2}FSI$ and $(PMpyr_f)_{0.5}Li_{0.5}FSI$ electrolytes. (i) C *1s* spectra, (ii) F *1s* spectra, (iii) N *1s* spectra, and (iv) atomic concentration.

anode cycling is measured in Cu/Li cells using Aurbach CE Protocol[37] with Gen 2, $(PMpyr_f)_{0.8}Li_{0.2}FSI$ and $(PMpyr_f)_{0.5}Li_{0.5}FSI$ electrolytes at current density of 0.1 mA/cm² (Fig. 5a–c). Gen 2 electrolyte consumes all Li inventory within only three cycles and displays a low CE of 45.0%. Both $PMpyr_fFSI$-based electrolytes demonstrated substantially higher CE, with $(PMpyr_f)_{0.5}Li_{0.5}FSI$ showing a higher CE of 97.9% compared to 96.5% for $(PMpyr_f)_{0.8}Li_{0.2}FSI$. Similar performance is also observed in long-term cycling of Li/Cu cells (Fig. 5d). Li was plated on to Cu foil for 6 h and stripped to 1 V for 100 cycles with 0.1 mA/cm² current density. Gen 2 electrolyte shows lower than 35% CE throughout 100 cycles, while $(PMpyr_f)_{0.8}Li_{0.2}FSI$ and $(PMpyr_f)_{0.5}Li_{0.5}FSI$ show high CE of 99% after initial stabilization. With decreased $PMpyr_f^+$ cation concentration, $(PMpyr_f)_{0.5}Li_{0.5}FSI$ shows a higher first cycle CE than $(PMpyr_f)_{0.8}Li_{0.2}FSI$, which suggests the SEI formed by $(PMpyr_f)_{0.5}Li_{0.5}FSI$ electrolyte is thinner and more efficiently protects the Li surface from further degradation.

Gen 2, $(PMpyr_f)_{0.8}Li_{0.2}FSI$ and $(PMpyr_f)_{0.5}Li_{0.5}FSI$ electrolytes were also examined in Li/Li symmetric cells (Fig. 5e and Supplementary Fig. S11). For Gen 2 electrolyte, after the initial stabilization, the overpotential continues to increase and starts to polarize after 700 h, which correlates to the undesired side reactions between the Gen 2 electrolyte and Li metal, which results in the accumulation of SEI layer, impedance buildup, and electrolyte depletion. The voltage profile also demonstrates a characteristic "peaking" behavior (Supplementary Fig. S11a) that is ascribed to different resistances of dendrite formation (nucleation), dendrite dissolution, bulk dissolution, and pitting from bulk Li surface[38]. This type of voltage trace is typically correlated to a dendritic morphology on a Li surface (Fig. 6a). For $(PMpyr_f)_{0.8}Li_{0.2}FSI$ electrolyte, the initial overpotential is slightly higher compared to $(PMpyr_f)_{0.5}Li_{0.5}FSI$ electrolyte, regardless of its lower viscosity, which suggests the excessive SEI formed by the fluorinated cation in $(PMpyr_f)_{0.8}Li_{0.2}FSI$ electrolyte has higher surface impedance. The overpotential continues decreases over cycling and cell experienced soft shorting issue after 600 h. This result is consistent with the higher portion of dendritic morphology observed in plated Li on Cu foil with $(PMpyr_f)_{0.8}Li_{0.2}FSI$ electrolyte (Fig. 6b). For $(PMpyr_f)_{0.5}Li_{0.5}FSI$ electrolyte, the voltage profile is stable over 900 h, which indicates that SEI formed by $(PMpyr_f)_{0.5}Li_{0.5}FSI$ sufficiently protects the Li-metal surface

and minimizes the undesired side reactions under the current testing condition. The stable overpotential also suggests that the Li surface area is relatively maintained during Li plating/stripping process, which could be due to the more densely packed columnar Li morphology in $(PMpyr_f)_{0.5}Li_{0.5}FSI$ electrolyte, as shown in Fig. 6c. The voltage profiles of $(PMpyr_f)_{0.8}Li_{0.2}FSI$ and $(PMpyr_f)_{0.5}Li_{0.5}FSI$ show the "arcing" behavior corresponding to a diffusion-controlled process (Supplementary Fig. S11b, c). The excellent Li-metal compatibility with $(PMpyr_f)_{0.5}Li_{0.5}FSI$ is further demonstrated by Li/Li symmetric cell using a thin Li foil of 20 μm. $(PMpyr_f)_{0.5}Li_{0.5}FSI$ exhibits stable voltage profile over 500 h testing, while Gen 2 polarizes within 50 h testing due to Li source being exhausted (Supplementary Fig. S11e, f). $PMpyr_fFSI$-based electrolytes show high Li-metal compatibility as demonstrated in both Li/Cu and Li/Li cells, which could be contributed by the SEI formed by both $PMpyr_f^+$ cation and FSI⁻ anion. Moreover, with the $PMpyr_f^+$ cation concentration optimization, the SEI formed by $(PMpyr_f)_{0.5}Li_{0.5}FSI$ electrolyte shows lower impedance, which enables uniform and highly densely packed Li morphology during Li plating.

**Protection of NMC622 surface**

Cycled NMC622 was recovered from the cycled half-cells for surface morphology and chemical structure analysis. High-resolution transmission electron microscopy (STEM) reveals a difference in the surface structure change (Fig. 7) with electrolytes. Previous studies have shown that, while the cathode degradation is mainly incurred by its intrinsic structural instability, the process usually starts at the cathode/electrolyte interface. Thus, a well-maintained layered surface structure after cycling confirms the efficacy of a protective CEI, which extensively suppresses the side reactions from propagating[39-41]. For NMC622 recovered from Gen 2 electrolyte, a thick surface rock-salt layer (~6 nm) is observed, with a 2-nm-thick CEI, and cracks inside the primary particles are observed. Such excessively thin CEI borders on the lower limit for interphases, as electron tunneling could easily occur. An electron energy loss spectroscopy (EELS) line scan was conducted from the primary particle surface to the bulk. The O-K pre-peak is much weaker at primary particle surface compared to bulk, suggesting more oxygen vacancies formed near surface. This result is

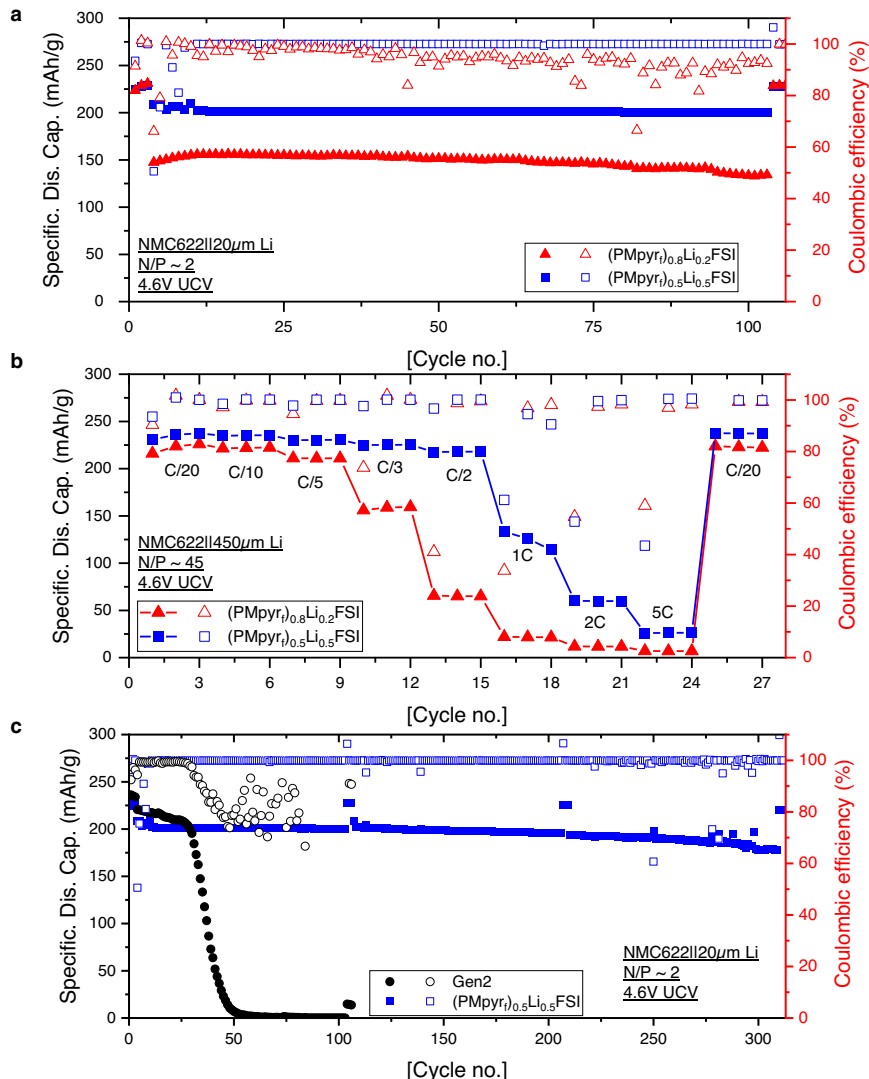

**Fig. 4 | NMC622/Li cell performance. a** Full-cell performance cycled using (PMpyr$_f$)$_{0.8}$Li$_{0.2}$FSI and (PMpyr$_f$)$_{0.5}$Li$_{0.5}$FSI electrolytes. **b** Rate capability of (PMpyr$_f$)$_{0.8}$Li$_{0.2}$FSI and (PMpyr$_f$)$_{0.5}$Li$_{0.5}$FSI electrolytes cycled in half-cell. **c** Long- term cycling performance for full cell using Gen 2 and (PMpyr$_f$)$_{0.5}$Li$_{0.5}$FSI electro- lytes. Cutoff voltage for all NMC622/Li cells is 4.6–3.0 V.

consistent with poor high-voltage performance in Gen 2 electrolyte and high surface impedance, as shown in Supplementary Fig. S9a. As the CEI formed by Gen 2 electrolyte cannot fully protect the NMC surface, electrolyte oxidation continues during high-voltage cycling, resulting in surface O loss and persistent phase transition from layer to resistive rock-salt structure[42–47]. For (PMpyr$_f$)$_{0.8}$Li$_{0.2}$FSI electrolyte, no inner granular cracking was observed in the primary particle even after 100 cycles. A thick CEI layer of 8 nm is observed, suggesting more electrolyte decomposition on the surface that provides sufficient protection against electron tunneling, but meanwhile introduces excessive interfacial impedance to Li$^+$-migration. The cation mixing layer observed on the surface is rather thin. The EELS line scan shows a very low signal at 5 nm depth correlating under this thick CEI. The pre- peaks observed from 10 nm are very similar to the bulk, suggesting that the CEI formed by (PMpyr$_f$)$_{0.8}$Li$_{0.2}$FSI is thick enough to prevent cathode surface degradation. The most striking difference is observed for the (PMpyr$_f$)$_{0.5}$Li$_{0.5}$FSI electrolyte cycled cathode. At decreased cation concentration, NMC cathode cycles only shows about a 2-nm surface transition layer (Fig. 7d), which is only 1 nm thicker compared to pristine NMC sample (Fig. 7a), while the interfacial impedance is much lower, as shown in Supplementary Fig. S9. No intra- or inter- granular cracking is observed, and after 300 cycles, the CEI is only

about 1 nm thick, and the EELS line scan shows similar pre-peaks from surface to bulk. These results indicate that the CEI formed by (PMpyr$_f$)$_{0.5}$Li$_{0.5}$FSI is fairly thin, while its chemistry renders it highly protective, which minimized the oxidation reaction on the electrolyte/ cathode interface, suppressed the O loss on the cathode surface, and preserved the layer structure. Apparently, the (PMpyr$_f$)$_{0.5}$Li$_{0.5}$FSI electrolyte achieved a proper balance of providing sufficient protec- tion of the cathode surface while minimizing resistance to ion transport.

In summary, we have demonstrated a fluorinated cation and its impact on interphasial chemistries for the first time in a high-voltage lithium-metal battery. The fluorination brings unique interfacial structure and subsequent interphasial chemistries on both Li$^0$ anode and high nickel cathode. By optimizing cation concentration in inner- Helmholtz layer, the strong electrode/cation surface interaction caused by fluorination can be attenuated and its optimal interphasial contribution can be achieved, which leads to high Coulombic effi- ciency, densely packed Li morphology, and excellent cycling stability of the 4.6 V lithium-metal battery. STEM demonstrated that a highly protective CEI formed by (PMpyr$_f$)$_{0.5}$Li$_{0.5}$FSI electrolyte maintains a thin cation mixing layer with minimum surface structure change. The new cation that carries fluorine will cast significant impact on

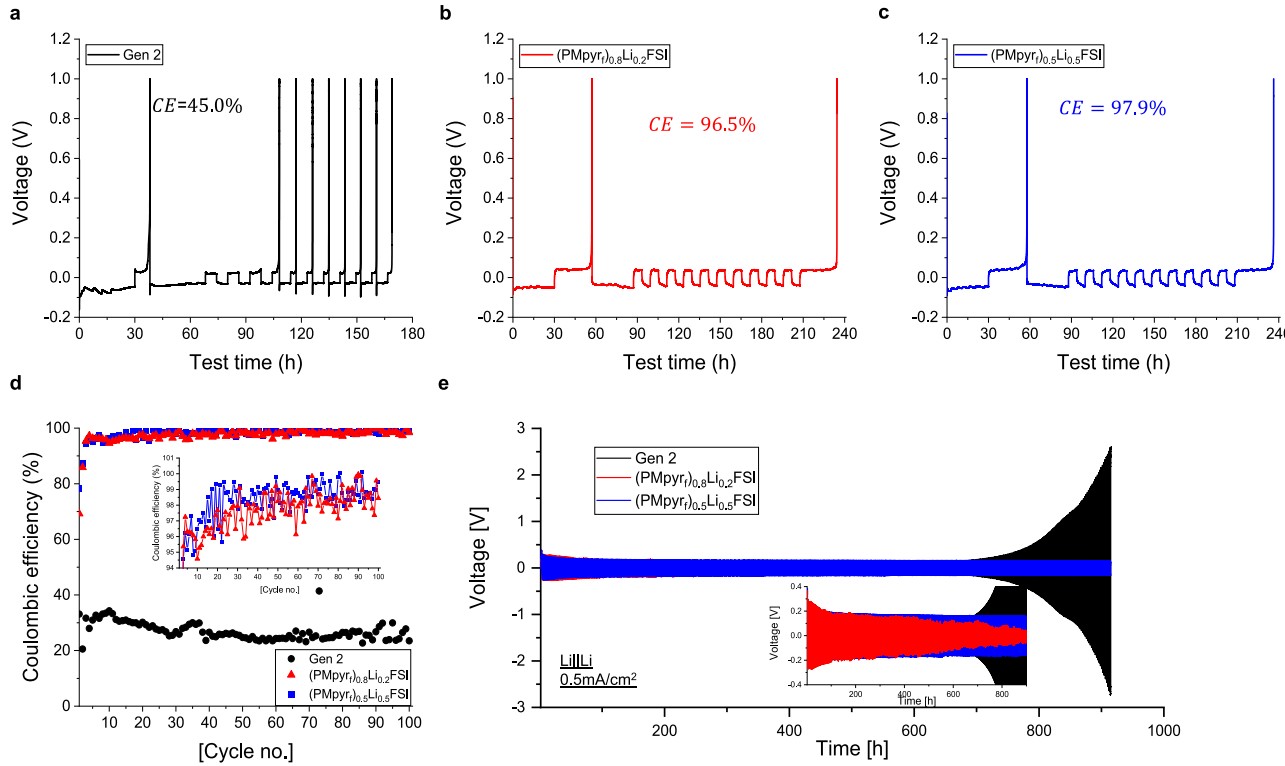

**Fig. 5 | Li/Cu and Li/Li symmetric cell performance. a**–**c** Voltage profile of an Aurbach test using **a** Gen 2, **b** $(PMpyr_f)_{0.8}Li_{0.2}FSI$, and **c** $(PMpyr_f)_{0.5}Li_{0.5}FSI$. **d** Coulombic efficiency of Li/Cu cell cycled with current density of 0.1 mA/cm². **e** Voltage profile of Li/Li symmetric cell with current density of 0.5 mA/cm².

emerging battery chemistries that desperately need tailored interphasial chemistries.

## Methods
### Electrode and electrolyte preparation
LiNi$_{0.6}$Mn$_{0.2}$Co$_{0.2}$O$_2$ cathode (90% NMC622, Toda Kogyo Corp.; 5% C45 conductive carbon, Timcal; 5% PVdF, Solvay 5130) was fabricated by the CAMP Facility at Argonne National Laboratory. Active material loading is 8.8 mg/cm²e. 1-Methyl-1-propyl-3-fluoropyrrolidinium bis(-fluorosulfonyl)imide (PMpyr$_f$FSI) was synthesized following the literature procedure[35] by the one-step reaction of 1-propyl-3-fluoropyrrolidine with MeFSI. The resultant ionic liquid was dried in a lyophilizer for at least two days, then stored in 4 Å molecular sieves and filtered before use. The water content was <20 ppm, measured by Karl–Fischer titrator C30. $(PMpyr_f)_{0.8}Li_{0.2}FSI$ and $(PMpyr_f)_{0.5}Li_{0.5}FSI$ electrolytes were prepared by dissolving LiFSI (Nippon ShokuBai Co., Ltd.) in PMpyr$_f$FSI with 4/1 and 1/1 molar ratio in an argon-filled glovebox.

### Physical and electrochemical properties
Electrochemical impedance spectroscopy (EIS) measurements were conducted on the Solartron Analytical 1400 Cell test station. Cell impedance was measured with the frequency range from 1 MHz to 0.1 Hz at the open circuit potential.

Galvanostatic charge-discharge cycling tests were conducted on the Maccor Electrochemical Analyzer (MIMSclient) with Al-coated 2032-coin cells. NMC622/Li cells were tested with a cutoff voltage of 4.6–3.0 V. Li/Li symmetric cells and Li/Cu cells were assembled using 450-µm-thick Li foil. The separator was a glass micro-fiber disc, and the total electrolyte amount is 100 µL. Cell testing was conducted at 30 ℃. For the C-rate test, C/20 was used as charging current, and different currents were used for discharge for three cycles. Aurbach test: 3 mAh/cm² Li reservoir is plated on the Cu foil using 0.1 mA/cm², then

0.6 mAh/cm² Li was stripped/plated for ten cycles before completely stripped to 1 V.

### X-ray photoelectron spectroscopy (XPS)
XPS analysis was conducted on a PHI 5000 VersaProbe II system (Physical Electronics) with a base pressure of ~2 × 10⁻⁹ torr. The spectra were obtained using an Al Kα radiation (hυ = 1486.6 eV) beam (100 µm, 25 W), with Ar⁺ and electron beam sample neutralization, in Fixed Analyzer Transmission mode with a pass energy of 11.75 eV. Subtracting a Shirley background and then fitting the spectra to multiple Gaussian peaks was performed on all spectra using the Multipack software from Physical Electronics. The area under the XPS peaks (the sum of the Gaussian components) was adjusted using manufacturer-calibrated relative sensitivity factors and normalized to obtain elemental concentrations. The same normalization factors were used to plot XPS signal intensities as concentration per unit energy (at % eV⁻¹). Binding energy was calibrated by shifting every region to align the C 1 s peak of C−C/C−H environments at 284.8 eV.

### Scanning electron microscopy and transmission electron microscopy (SEM/TEM)
The cycled coin cells were disassembled in the argon-filled glovebox, and the electrodes were thoroughly rinsed with anhydrous dimethyl carbonate and allowed to air-dry. The morphologies and the elemental mapping of the cycled electrodes were examined using SEM and energy-dispersive X-ray spectroscopy (EDS) using the JOEL JCM-6000-PLUS. TEM analysis was conducted in the JOEL JEM-2100F.

### Density functional theory methods
Periodic spin-polarized DFT calculations were performed using the plane-wave-based Vienna ab initio Simulation Package (VASP)[48]. The projectoraugmented wave (PAW) method[49] was used to describe the

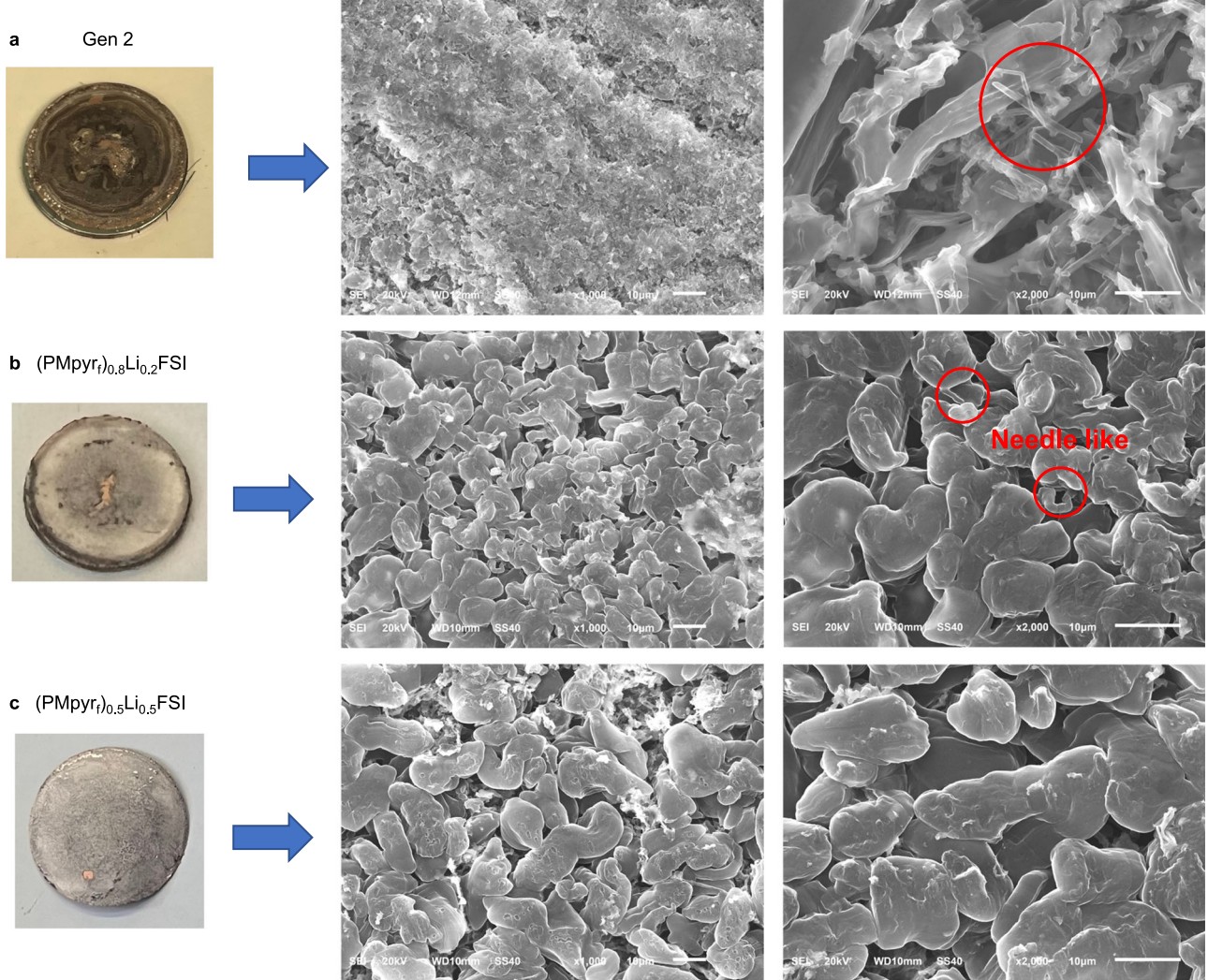

**Fig. 6 | SEM of plated Li on Cu foil with current density of 0.1 mA/cm².** **a** Gen 2, **b** (PMpyr$_f$)$_{0.8}$Li$_{0.2}$FSI, and **c** (PMpyr$_f$)$_{0.5}$Li$_{0.5}$FSI electrolytes.

wave functions of the ionic cores; and the generalized gradient approximation (GGA) PBE functional was used to account for the electron exchange-correlation effects[50]. Energy cutoff of 520 eV and 440 eV was used to optimize the bulk for the Li anode and NMC cathode, respectively, and the corresponding Monkhorst-Pack $k$-point meshes were $8 \times 8 \times 8$ and $3 \times 3 \times 3$[51]. The periodic four layers of surface slab of Li (110) and NMC622 (100) was created using materials studio, and the length of the vacuum along the $z$-direction is at least 20 Å to avoid spurious interactions between the periodic images. For NMC622 cathode surface calculations, 400 eV energy cutoff with a single Γ-point was used, while a $k$-point mesh of $3 \times 2 \times 1$ was used for Li anode system with the same energy cutoff. In order to correct the d-electron delocalization for Ni, Mn, and Co, the Hubbard U (GGA + U) was applied[52]. The U values for Ni, Mn, and Co are 6.34, 4.48, and 5.14 eV, respectively, which were obtained from previous works[53]. The energy and geometries of the gas phase species were calculated by placing the molecule in a box with dimensions of $25 \times 25 \times 25$ Å³. A single Γ-point was used for these calculations. For all calculations, the convergence criterion for the self-consistent iteration is $1 \times 10^{-5}$ eV; and the ionic relaxations stop when the force on each atom is less than 0.05 eV/Å.

To evaluate the reducibility of the cation and anion of the IL with and without F-substitution, reduction potential was computed using Eq. (1)[54], where $n$ is the number of electrons transferred during reduction and $\triangle G^{red}$ is the reaction free energy of the reduction, which

is approximately equal to the $\triangle E$ from DFT, since the reactants are strongly adsorbed and we anticipate the $\triangle S$ would be negligible. $E_H$ and $E_{Li}$ represents the reduction potentials of the absolute hydrogen electrode and Lithium electrode, which are 4.44 V and −3.05 V, respectively.

$$E_{wrt\ Li}^0 (V) = - \left( \frac{\triangle G^{red}}{n} + E_H \right) - E_{Li} \tag{1}$$

In order to explore the electronic properties of the ionic liquid with and without F-substitution, Gaussian calculations were carried out using Gaussian 09[55] simulation package with B3LYP functional and TZVP basis set[56,57].

**Molecular dynamics (MD) simulations**

All MD simulations were performed on the Cray cluster Theta of the Leadership Computing Facility at Argonne National Laboratory. The simulations were carried out in a high-performance mode with version 2.14 of NAMD[58], a greatly scalable molecular dynamic program used to render an atom-by-atom representation of biomolecules, created at the University of Illinois. All electrolyte simulations adopted GAMMP force field[59] generated through GAAMP server on Laboratory Computing Resource Center at Argonne National Laboratory. The molar ratios of each molecular species, Li⁺, FSI⁻, and PMpyr⁺ (or PMpyr$_f^+$) match the ratio in (PMpyr)$_{0.8}$Li$_{0.2}$FSI, (PMpyr$_f$)$_{0.8}$Li$_{0.2}$FSI, and (PMpyr$_f$)$_{0.5}$Li$_{0.5}$FSI

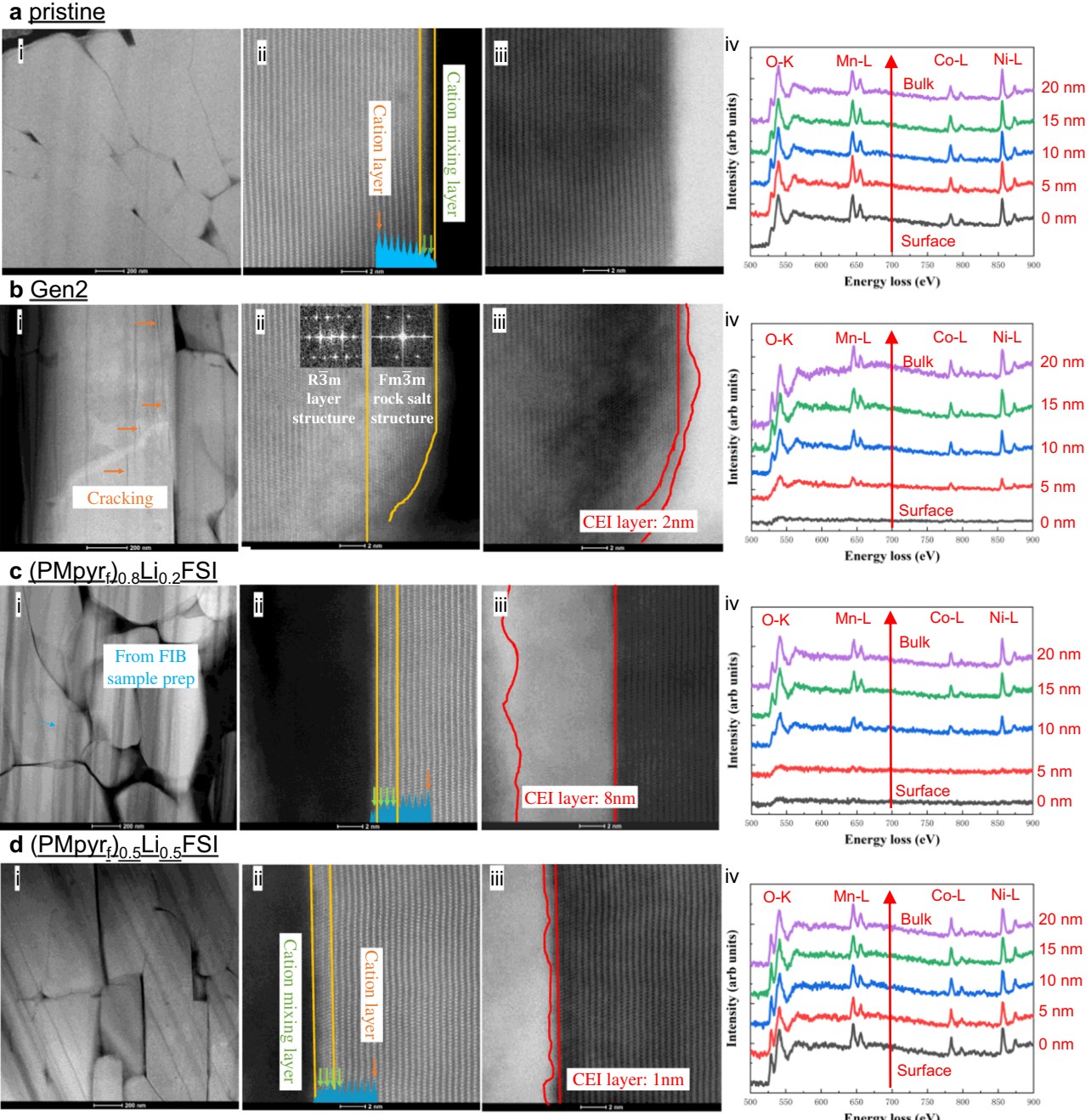

**Fig. 7 | HAADF-STEM/EELS analysis of cycled NMC622 cathodes. a** Pristine NMC622 and cycled NMC622 cathodes with different electrolytes, **b** Gen 2 electrolyte after 100 cycles, **c** (PMpyr$_f$)$_{0.8}$Li$_{0.2}$FSI electrolyte after 100 cycles, and **d** (PMpyr$_f$)$_{0.5}$Li$_{0.5}$FSI electrolyte after 300 cycles. (i–ii) HAADF-STEM images, (iii) BF-STEM images, and (iv) EELS line scan from primary particle surface to bulk of cathode.

electrolytes. The composite system of the NMC622 cathode, electrolyte, and Li-metal anode are constructed and modeled with all-atom models reported in our previous study[60]. The number of atoms for the simulated systems range between 418,300 and 418,600, and dimension of each simulation cell is ~100 Å × 100 Å × 560 Å, including a 125 Å vacuum between the anode and image of the cathode to eliminate spurious long-range interactions. All simulations adopt Periodic Boundary Condition under constant NVT condition. The equations of motion were integrated with a 2 fs time step, using Langevin dynamics at a temperature of 300 K. To overcome the slow dynamics due to the high viscosity of the simulated systems, Hamiltonian simulated annealing method (HSA)[61] was employed to accelerate the equilibration of MD trajectories. In each HSA simulation, 64 independent trajectories were generated, and all of them were used to get a statistical average of structural properties. Each trajectory lasts 20 ns and the last 10 ns was adopted to do a statistical average. Snapshot of each trajectory was sampled at an interval of 100 ps, and thus for each simulated system, 6400 snapshots were adopted to guarantee high-fidelity sampling.

## Data availability
Source data are provided with this paper.

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

## Acknowledgements

This work was funded by the U.S. Department of Energy (DOE), Office of Energy Efficiency and Renewable Energy, Vehicle Technologies Office. Support from Tien Duong, Peter Faguy and Mallory Clites of Vehicle Technology Office is gratefully acknowledged. This research used the Leadership Computing Facility resource at Argonne National Laboratory, which is supported by DOE Office of Science under Contract No. DE-AC02-06CH11357. TEM/EELS were performed using EMSL (grid.436923.9), a DOE Office of Science User Facility sponsored by the Office of Biological and Environmental Research located at Pacific Northwest National Laboratory (PNNL). The submitted manuscript has been created by UChicago Argonne, LLC, Operator of Argonne National Laboratory.

## Author contributions

Z.Z. and K.X. conceived the idea and proposed the research. Q.L. designed experiments, performed the electrochemical measurements, characterized materials and analyzed the data. K.P. synthesized the MeFSI. W.J. performed the molecular dynamics simulations. J.X., D.-J.Y., and C.L. performed the DFT calculations. Y.X. and C.W. performed the TEM/EELS analysis. Z.Y. performed the XPS measurement. K.X. and Z.Z. contributed to the discussion and provided suggestions. Q.L., W.J., K.X., and Z.Z. wrote the manuscript with inputs from all other co-authors.

## Competing interests

The authors declare no competing interests.
