## [Peer Review File · Nature Communications]

REVIEWER COMMENTS

Reviewer #1 (Remarks to the Author):

The authors have addressed the points raised in the earlier comments

Reviewer #2 (Remarks to the Author):

The Reviewer appreciates the authors' effort to address the previous comments. However, the comments have not been addressed well, for example, the XPS analysis after in-depth etching was not conducted, thus the conclusion drawn was not convincing. One major concern is how to distinguish the contribution of SEI from F-substituted cation or FSI anion. According to the XPS of SEI, it seems that after increasing the cation concentration, the contents of C-F (F 1s) and C-N+ (N 1s) increased, and the authors concluded that the contribution of cation is increased. However, if we check the results more carefully, the content of LiF (F 1s) reduced and the FSI (N 1s) increased. These results seem contradictory, what's the decomposed products of F-substituted cation and FSI anion? What's the components in SEI? It also brings new questions, did the authors have completely cleaned the surface? Is there any salt residue on the surface? To understand the components of SEI, in-depth etching XPS is required. Another question is why glass fiber was used as separator, not the commonly used Celgard separator.

Reviewer #3 (Remarks to the Author):

In the manuscript, Zhang and co-workers synthesize a cation that carries fluorine in order to pre-store the fluorine source on positive-charged species. Using the ionic liquid electrolyte, the authors explore the influence of the electrolyte on Li metal morphology, coulombic efficiency, cycling performances, and its interface with an NMC622 cathode.

Generally, the research presents some interesting results. However, some issues should be addressed before further consideration can be taken.

1. The authors wrote in the manuscript that “With F-substitution, PMpyrf⁺ is more active towards reduction, almost as active as FSI⁻ anion.” Since the DFT data show there is a stronger driving force for FSI reduction over the new cation, wouldn't the SEI still be anion dominated? In addition, according to the authors' description, the reduction potential of FSI⁻ (5.43 V) is higher than that of PMpyrf⁺ (5.16 V), indicating that the contribution rate of anion FSI⁻ to SEI is higher.

2. Although ionic electrolyte has some advantages, its disadvantages such as high cost, low conductivity and high viscosity are also significant. As shown in Figure 4b, the electrochemical performance is weak at high current rate (1 C). Therefore, the authors should provide the cycling performance at relatively high current rates (e.g, 1C), which is very important to evaluate the performance of the electrolyte.

3. The value of Coulombic efficiency for Li plating/stripping should be compared with those reported for similar electrolytes.

Response to Reviewers Comments

Reviewer #2 (Remarks to the Author):

The Reviewer appreciates the authors' effort to address the previous comments. However, the comments have not been addressed well, for example, the XPS analysis after in-depth etching was not conducted, thus the conclusion drawn was not convincing. One major concern is how to distinguish the contribution of SEI from F-substituted cation or FSI anion. According to the XPS of SEI, it seems that after increasing the cation concentration, the contents of C-F (F 1s) and C-N+ (N 1s) increased, and the authors concluded that the contribution of cation is increased. However, if we check the results more carefully, the content of LiF (F 1s) reduced and the FSI (N 1s) increased. These results seem contradictory, what's the decomposed products of F-substituted cation and FSI anion? What's the components in SEI? It also brings new questions, did the authors have completely cleaned the surface? Is there any salt residue on the surface? To understand the components of SEI, in-depth etching XPS is required. Another question is why glass fiber was used as separator, not the commonly used Celgard separator.

Reply:

(1) Thanks for this reviewer's diligent review and excellent comments. Considering all the questions this reviewer raised, we assembled new cells and conducted the surface and in-depth XPS analysis of the cycled Li metal anodes. All the XPS samples were thoroughly rinsed by anhydrous DMC solvent to make sure there is no residue salt on the surface of the samples.

(2) The surface XPS analysis and in-depth XPS profiling for cycled lithium anode are provided below in Figure R1 and Figure R2. The peak at 399 eV in the N1s spectra was misassigned to FSI anion decomposition product. The new assignment should be LiNC_xH_y species according to literature¹, which is the ring-opening product from F-cation decomposition as indicated by DFT calculation. Therefore, there is no contradiction in our results. The F-cation prefers to undergo defluorination followed by ring-opening reaction, which leads to the formation of LiNC_xH_y and LiF. The deprotonation pathway is also possible which will lead to C-F component in the SEI. But this mechanism is less favorable. Total peak intensity for C-N⁺ + LiNC_xH_y as F-cation decomposition products is higher for $(\text{PMpyr}_f)_{0.8}\text{Li}_{0.2}\text{FSI}$ cycled anode than that for $(\text{PMpyr}_f)_{0.5}\text{Li}_{0.5}\text{FSI}$, indicating cation contribution to SEI can be tuned by varying its concentration (from $(\text{PMpyr}_f)_{0.8}\text{Li}_{0.2}\text{FSI}$ sample to $(\text{PMpyr}_f)_{0.5}\text{Li}_{0.5}\text{FSI}$ sample). The optimal cation and anion contribution to the SEI is found for the $(\text{PMpyr}_f)_{0.5}\text{Li}_{0.5}\text{FSI}$ sample as indicated by Figure R1.

Atomic percentage data further confirm this conclusion. As can be seen from Figure R3, with high F-cation concentration ($(\text{PMpyr}_f)_{0.8}\text{Li}_{0.2}\text{FSI}$ sample), the C atomic concentration is higher, and S is lower compared to $(\text{PMpyr}_f)_{0.5}\text{Li}_{0.5}\text{FSI}$ sample on the surface and at all depths. Since S is mainly contributed by FSI anion, we conclude the relative cation contribution to SEI is increased with increasing cation concentration.

With increasing depth from surface, we observe that the inorganic LiNC_xH_y (cation), LiF (cation + anion) and $\text{SO}_2\text{F-N}$ (anion) species dominate the SEI, suggesting both cation and FSI anion contribute to the inner layer of SEI and F-cation mainly contributes through the defluorination pathway. This is aligned well with our hypothesis that the F-cation populates the inner Helmholtz layer and engenders higher fluorination at the interfaces.

(3) Fluorinated ionic liquid electrolyte $(\text{PMpyr}_f)_{0.5}\text{Li}_{0.5}\text{FSI}$ is highly hydrophobic; they cannot wet the standard Celgard PP/PE separators (Celgard 2500/Celgard 2325). Therefore, glass fiber separator has been used for the battery performance demonstration.

Figure R1. XPS surface analysis of cycled Li metal anode (also provided in the revised manuscript).

Figure R2. In-depth XPS profiling of Li metal anode cycled with $(\text{PMpyr}_f)_{0.8}\text{Li}_{0.2}\text{FSI}$ and $(\text{PMpyr}_f)_{0.5}\text{Li}_{0.5}\text{FSI}$.

Figure R3. SEI component atomic concentration on the cycled Li metal at different depths.

Reviewer #3 (Remarks to the Author):

In the manuscript, Zhang and co-workers synthesize a cation that carries fluorine in order to pre-store the fluorine source on positive-charged species. Using the ionic liquid electrolyte, the authors explore the influence of the electrolyte on Li metal morphology, coulombic efficiency, cycling performances, and its interface with an NMC622 cathode.

Generally, the research presents some interesting results. However, some issues should be addressed before further consideration can be taken.

1. The authors wrote in the manuscript that “With F-substitution, PMpyrf⁺ is more active towards reduction, almost as active as FSI⁻ anion.” Since the DFT data show there is a stronger driving force for FSI reduction over the new cation, wouldn't the SEI still be anion dominated? In addition, according to the authors' description, the reduction potential of FSI⁻ (5.43 V) is higher than that of PMpyrf⁺ (5.16 V), indicating that the contribution rate of anion FSI⁻ to SEI is higher.

Reply:

Thanks for reviewer 3's encouraging review on our work. We agree the original statement is not clear, and we have reworded it to “With F-substitution, PMpyrf⁺ is more active towards reduction compared to PMpyr⁺ and is almost as active as FSI⁻ anion.” The more active towards reduction is actually a comparison between PMpyrf⁺ and PMpyr⁺; and the reduction potential for PMpyrf⁺ is lower than, but close to FSI, that's what we meant by “almost as active as FSI”. Although there is a difference between FSI and PMpyrf⁺ in reduction potential, the value is small. As we can see from the XPS results, both FSI anion and the F-cation contributes to the SEI formation. The key is to find the optimal relative contribution from both anion and cation. By varying the cation concentration, we are able to adjust the relative contribution of cation and anion on the SEI. The optimal result is achieved by the (PMpyrf)_{0.5}Li_{0.5}FSI sample. In the XPS spectra for this sample, the relative peak intensity of SO₂F-N (anion) and C-N⁺/LiNC_xH_y (cation) are comparable in N1s spectra (Figure R1). Therefore, FSI contribution to SEI is not dominate and both cation and anion contribution are critical to the SEI.

2. Although ionic electrolyte has some advantages, its disadvantages such as high cost, low conductivity and high viscosity are also significant. As shown in Figure 4b, the electrochemical performance is weak at high current rate (1 C). Therefore, the authors should provide the cycling performance at relatively high current rates (e.g, 1C), which is very important to evaluate the performance of the electrolyte.

Reply:

Thanks for the comments. The cost for the new PMpyrFSl ionic liquid is much lower because a one-step reaction of 1-propyl-3-fluoropyrrolidine with MeFSI has been employed. Traditional ionic liquid synthesis involves two-step reaction with relatively low yield and low purity, requiring extensive purification and raising the cost. As the reviewer pointed out, for this new developed ionic liquid electrolyte, it still has the limitation for high-rate performance. For the current work, our focus is on the molecular design and its corresponding impact on interphasial stability. To improve its power capability, we have been working on new strategies such as addition of new diluent co-solvents to reduce the viscosity and enhance the conductivity. The manuscript has been written up and will be published in a separate work.

Figure R4 shows the cycling data at 1 C rate. As expected, the capacity is relatively low at high current due to the high viscosity and low conductivity, but the capacity is fully recovered after 200 cycles at low rate C/20 (223 mAh/g at the 206th cycle).

Figure R4. NMC622/Li cell performance with (PMpyr)_{0.5}Li_{0.5}FSI electrolyte cycled between 4.6-3.0 V. Three formation cycles at C/20, then cycled at 1 C for 200 cycles and 3 cycles at C/20 in the end for capacity check.

3. The value of Coulombic efficiency for Li plating/stripping should be compared with those reported for similar electrolytes.

Reply:

That is a good point. Following this reviewer's comments, we have added a comparison **Table S1** in the Supporting Information. For 3.2 mol/kg LiFSI in C3mpyrFSI IL, the reported CE is 85% with 0.5mA/cm²; for LiFSI:EmimFSI=1/2, CE is 98.22%. The CE for our (PMpyr_f)_{0.8}Li_{0.2}FSI and (PMpyr_f)_{0.5}Li_{0.5}FSI is 96.5% and 97.9%, respectively with 0.1 mA/cm² current and 3 mAh/cm² Li reservoir. Since our fluorinated cation populates in the inner-Helmholtz layer and has a strong interaction with Li metal, when tested at higher current, voltage polarization to 5 V is observed in the middle of the test due to Li⁺ ion depletion and the target Li reservoir (3 mAh/cm²) cannot be reached. Therefore, a lower current was chosen for this Cu/Li CE test.

Table S1. Comparison of Coulombic efficiency of literature reported IL electrolytes and this work in Li/Cu cells.

Electrolyte	Test Current (mA/cm ²)	Li Reservoir (mAh/cm ²)	Coulombic Efficiency (%)
[LiFSI] ₁ [EmimFSI] ₂ ¹	0.5	2.5	98.2
3.2 mol/kg LiFSI in C3mpyrFSI ²	0.5	4.0	85.0
[LiFSI] ₁ [Pyr ₁₄ FSI] ₄ or [LiFSI] ₃ [Pyr ₁₄ FSI] ₄ ³	0.5	1.0	NA (short circuit)
(PMpyr _f) _{0.8} Li _{0.2} FSI ^{this work}	0.1	3	96.5%
(PMpyr _f) _{0.5} Li _{0.5} FSI ^{this work}	0.1	3	97.9%

References:

1. Liu, X.; Mariani, A.; Diemant, T.; Pietro, M. E. D.; Dong, X.; Kuenzel, M.; Mele, A.; Passerini, S., Difluorobenzene-Based Locally Concentrated Ionic Liquid Electrolyte Enabling Stable Cycling of Lithium Metal Batteries with Nickel-Rich Cathode. *Adv. Energy Mater.* **2022**, *12* (25), 2200862.
2. Pal, U.; Rakov, D.; Lu, B.; Sayahpour, B.; Chen, F.; Roy, B.; MacFarlane, D. R.; Armand, M.; Howlett, P. C.; Meng, Y. S.; Forsyth, M., Interphase Control for High Performance Lithium Metal Batteries Using Ether Aided Ionic Liquid Electrolyte. *Energy Environ. Sci.* **2022**.
3. Liu, X.; Zarrabeitia, M.; Mariani, A.; Gao, X.; Schütz, H. M.; Fang, S.; Bizien, T.; Elia, G. A.; Passerini, S., Enhanced Li⁺ Transport in Ionic Liquid-Based Electrolytes Aided by Fluorinated Ethers for Highly Efficient Lithium Metal Batteries with Improved Rate Capability. *Small Methods* **2021**, *5* (7), 2100168.